# Optimizing test and treat options for vivax malaria: An options assessment toolkit (OAT) for Asia Pacific national malaria control programs

Sanjaya Acharya[1], Manash Shrestha[2,3], Ngo Duc Thang[4], Lyndes Wini[5], M. Naeem Habib[6], Josselyn Neukom[7], Karma Lhazeen[2], Caroline A. Lynch[1,2,3‡], Kamala Thriemer[1‡]*

1 Global Health Division, Menzies School of Health Research and Charles Darwin University, Darwin, Australia, 2 Asia Pacific Malaria Elimination Network (APMEN) Vivax Working Group, Singapore, Singapore, 3 MMV Medicines for Malaria Venture (MMV), Geneva, Switzerland, 4 National Institute of Malariology, Parasitology and Entomology (NIMPE), Ministry of Health, Hanoi, Viet Nam, 5 National Vector Borne Disease Control Programme, Ministry of Health and Medical Services, Honiara, The Solomon Islands, 6 National Malaria Control Program, Ministry of Public Health, Kabul, Afghanistan, 7 APLMA Consultant, Ho Chi Minh City, Viet Nam

‡ CAL and KT are shared last authors on this work.
* Kamala.Ley-Thriemer@menzies.edu.au

**Data Availability Statement:** Individual data cannot be shared publicly because of the risk that

## Abstract

Designing policy in public health is a complex process requiring decision making that incorporates available evidence and is suitable to a country's epidemiological and health system context. The main objective of this study was to develop an options assessment toolkit (OAT) to provide a pragmatic and evidence-based approach to the development of policies for the radical cure (prevention of relapse) of vivax malaria for national malaria control programs in the Asia-Pacific region. The OAT was developed using participatory research methods and a Delphi process using a sequential multi-phase design, adapted with a *pre-development phase*, a *development phase*, and a *final development phase*. In the *pre-development phase*, a literature review was conducted to inform the toolkit development. Data collection in the *development phase* consisted of core research team discussions, multiple rounds of consultation with participants from National Malaria Control Programs (NMP) (online and in person), and two separate modified e-Delphi processes with experts. The *final development* phase was the piloting of the toolkit during the annual meeting of the Asia Pacific Malaria Elimination Network (APMEN) Vivax Working Group. We developed a tool kit containing the following elements: i) Baseline Assessment Tool (BAT) to assess the readiness of NMPs for new or improved coverage of radical cure, ii) eight scenarios representative of Asia Pacific region, iii) matching test and treat options based on available options for G6PD testing and radical cure for the given scenarios, iv) an approaches tool to allow NMPs to visualize considerations for policy change process and different implementation strategies/approaches for each test and treat option. The OAT can support vivax radical cure policy formulation among NMPs and stakeholders tailoring for their unique country context.

respondents could be identified. Given the context of the interviews with NMPs even with identifying information removed, it is relatively easy to identify respondents. The same is the case for the expert replies in the Delphi process, given the large amount of open-ended free text answers. However, data are available on reasonable request via emailing to ethics@menzies.edu.au.

**Funding:** Funding for this work was provided by MMV and APLMA through the VivAccess and APMEN grants. The work was supported, in whole or in part, by the Bill & Melinda Gates Foundation [OPP1194815 / INV-008202]. Under the grant conditions of the Foundation, a Creative Commons Attribution 4.0 Generic License has already been assigned to the Author Accepted Manuscript version that might arise from this submission. The funders had no role in the study design, data collection and analysis, decision to publish, or preparation of the manuscript.

**Competing interests:** The authors have declared that no competing interests exist.

Future studies are needed to assess the utility and practicality of using the OAT for specific country context.

## Introduction

The aim of policy formulation for public health is to identify the best, often most cost-effective policies that will have a significant impact on a population's health. However, the process of policy making is complex requiring decision-making processes that rely on knowledge of a specific health problem, consideration and interpretation of the evidence base of potential interventions and an understanding of their feasibility and acceptability in a given country context [1]. Formulating a new policy is seldom a linear process but rather iterative and requires the incorporation of opinion and feedback of experts and multiple stakeholders [2]. At the outset, the evidence related to the topic itself requires critical interpretations followed by intersectoral coordination. To further complicate the process, policy decisions are sometimes made amidst limited available evidence [3, 4].

Policymakers often have to navigate the trade-offs between benefits at individual, social and national level against available resources. Balancing resources or their utilization can have a pivotal role for decision making and attests to the feasibility and practicality of the policy. In addition, addressing equity concerns, balancing political influence, and interests, also affect how a policy may be prioritized and formulated. Finally, a policy needs to be scrutinized for its public relevance and acceptability. A policy that has received widespread approval from policymakers could ultimately face criticism if it fails to foresee and address implementation challenges [5].

In order to facilitate evidence-based policymaking, various support mechanisms such as systematic reviews, evidence briefs, policy briefs, and stakeholders meeting reports are often used to gather information about the topic of interest (e.g. disease, epidemiology), the context (background, local, social and cultural context), evidence around the interventions, and where the policy will be implemented [6]. Evidence based policy formulation is often affected by a myriad of barriers such as limited political will, deficits in relevant research, and limitations in available resources [7].

Echoing the complexity of the policy making process for public health measures, malaria related policy formulation and especially vivax malaria incurs further specific challenges. The management of vivax malaria requires developing policies that address the complexity of the disease. Vivax malaria is associated with significant morbidity and mortality [8], and its control is more challenging than that of falciparum malaria. This is mainly because of important biological differences such as the ability of the parasite to form dormant liver forms that can reactivate weeks or months after the initial infection causing recurrent disease (relapse) and contributing to transmission. Radical cure, the treatment of both blood and liver stages of the parasite, is therefore crucial for reducing the vivax malaria disease burden [9]. The only widely available drug to kill the liver forms of the parasite and thereby preventing relapse is primaquine. However, its use has been limited by safety concerns, specifically among patients with reduced activity of the enzyme Glucose-6-Phosphatase Dehydrogenase (G6PD), and thus G6PD testing is recommended prior to their use [10]. Furthermore, effectiveness of primaquine is affected by its long treatment course resulting in low adherence and suboptimal dosing regimens [11]. Recent advances, including higher-dose shorter course [11–13] and ultra-short course primaquine [14], single dose tafenoquine as a new drug [15, 16], and novel

quantitative G6PD Point-of-care (POC) testing devices [17], are changing the landscape of products available for national malaria programs (NMPs) to tackle vivax malaria, but they haven't been incorporated into global or national guidelines yet.

Incorporating new evidence into malaria treatment policy is challenging because of the resources, expertise and time required to review and synthesize available evidence. Attention is directed towards the necessity for high-quality evidence, transparent guidance, and involving stakeholders in the decision-making process [18]. In the context of malaria, evidence review is usually done at the global level and this guidance is then translated into more specific national treatment guidelines. A previous seven-country comparison showed that time taken for this national malaria policy change varied from three months to three years in the Asia Pacific [19]. It is likely that future global guidelines will provide guidance on multiple radical cure options, and countries will need to identify the most suitable options for their context. This increases the complexity of decision making at national level, likely prolonging timelines for national adoption [20]. However, shortening the timelines for policy change is pivotal to meet proposed malaria elimination milestones of 2030 and country-level preparedness is therefore key to accelerating the policy decision process. NMPs in the Asia Pacific region have identified the need for more support to make informed decisions for optimal new radical cure options as they become available [21].

The aim of this work was therefore to develop and test an options assessment toolkit (OAT) which provides a practical and evidence-based approach for national decision-makers to determine optimal radical cure options for their given context.

## Material and methods

### Study design

The methodology used for the development of OAT has been described in detail previously [22]. In brief, we used participatory research methods with co-developing NMP participants and regional experts. A sequential multi-phase design was adapted with i) a *pre-development phase*, ii) a *development phase*, and iii) a *final development phase*.

In the *pre-development phase*, a literature review was conducted with the aim to identify reviews and descriptions of public health toolkits or other toolkits developed for health policy makers. This provided the basis for the *development phase* during which data for the OAT was collected. The *final development phase* was the piloting where the OAT tools were introduced to a larger group of NMP participants during the Asia Pacific Malaria Elimination Network (APMEN) Vivax Working Group annual meeting, held in Bangkok, in December 2022. Feedback from the meeting informed the final version of the OAT.

### Data collection

Data collection and development of the tools was conducted in the *development phase* and consisted of i) core research team discussions, ii) multiple rounds of consultation with NMPs participants (online and in person), and iii) and two separate modified e-Delphi processes with experts.

Internal team discussions were held weekly to formulate ideas, discuss feedback from NMP participants and experts and develop the toolkit.

Consultations were conducted with NMP participants from Afghanistan (MNH and team, 3 events), Solomon Islands (LW and team, 3 events), and Vietnam (NDT and team, 3 events) either as online or in person in-depth interviews by two researchers (JN and SA). Each interview with NMP participants lasted 60–90 minutes. One in-person working group interview was conducted in Bangkok where the NMP participant from Vietnam attended in person and

other NMP participants attended virtually. Online interviews were recorded after obtaining verbal consent and notes were transcribed and subjected to thematic analysis. The feedback of NMP participants, their contextual suggestions, and insights were used for validation and refinement of the OAT tools.

Regional experts were invited for two separate modified e-Delphi processes, each with multiple rounds. The modified e-Delphi process was used to improve the response rate and provide a solid grounding in previously developed work [23]. Consensus was defined when ≥75% of experts agreed on an answer to a question. Only those items that failed to reach consensus and any new items added by the experts through open-ended questions during the first round were moved into consecutive rounds. An online platform- *paperform.co* was used to collect the responses from the experts and weekly reminder emails were sent to non-responsive experts for at least two weeks.

## Toolkit development team

The initial core team comprised of six members who are either active researchers in the field of malaria (SA, MS, CAL, KT) or are involved in malaria programming and policy (JN, KL). The NMP participants were selected as outlined previously [22]. NMP participants included policymakers from the National Malaria and Leishmaniasis Control Program, Afghanistan (MNH), the National Vector Borne Disease Control Program of the Solomon Islands (LW), and the National Institute of Malariology, Parasitology and Entomology (NIMPE) in Vietnam (NTD).

## Experts for Delphi process

The detailed criteria for expert selection to participate in the modified e-Delphi process have been described previously [22]. In brief, experts had more than 10 years of professional experience in malaria research, a high-level expertise in malaria demonstrated by research publications on vivax malaria and/or health system strengthening, as well as capacity and willingness to contribute. Gender balance, and representation from different countries within the Asia Pacific region was considered.

A total of 32 regional experts were approached for the first modified e-Delphi process, and 21 (66%) participated. A majority of experts who participated in the e-Delphi process were from South-East Asia (12, 57%), had an academic/research background (19, 90%), with specific expertise in vivax diagnosis-treatment and malaria epidemiology (16, 76%) (S1 Table). A total of five rounds were conducted.

In the second modified e-Delphi process, there were 22 expert respondents out of the originally contacted and invited experts in the first process. A total of three rounds were conducted in this second process. Throughout the multiple rounds of e-Delphi, the response rate remained above 90%, and only dropped to 64% (i.e., 14/22) in round three of the second e-Delphi process.

**Ethical approval.**   Experts provided written informed consent to participate in the Delphi-process. All research was done in in accordance with the Declaration of Helsinki and the study protocol was approved by the Human Research Ethics Committee of the Northern Territory Department of Health and Menzies School of Health Research (HREC #22–4245).

## Results

### Overview of toolkit

The iterative and interactive OAT development process with the NMP participants resulted in some deviations from the originally proposed toolkit composition. For example, in the

**Table 1. Tools included in the OAT.**

| Tools of OAT | Objective |
|---|---|
| 1. Baseline Assessment Tool (BAT) | To assess the readiness of NMPs for new or improved coverage of radical cure |
| 2. A range of scenarios representative of Asia Pacific region | To aid NMPs in viewing different contextual and health system features of scenarios of malaria program phases in the region |
| 3. Test and treat options (based on available options of G6PD testing and radical cure) | To enable NMPs assess combinations of radical cure options for their current scenario |
| 4. Approaches tool (S1 Text) | To allow NMPs to visualize considerations for policy change process and different implementation strategies/approaches for each test and treat option |
| 5. Step-by-step user guide (S2 Text) | To provide specific instructions to use the OAT |

protocol [22], the development of an algorithm or decision tree was proposed originally, but NMP participants considered an approach where scenarios are matched to different radical cure options to be more user-friendly. Similarly, the proposed weighting tools were removed, because of concerns over increased complexity and reduced user friendliness (S2 Table). Tools included in the final OAT are listed in Table 1.

## Baseline Assessment Template (BAT)

The Baseline Assessment Template (BAT) was designed as a tool to assess the readiness of countries to implement new or improved radical cure options. It contains three domains relevant for policy change decision making: epidemiology, implementation capacity, and enabling environment. Each domain includes a set of factors, a question to capture this factor, and categories that quantify or contextualize each factor.

**Factors included in the BAT.** Overall, the BAT includes 14 factors, with five factors in the epidemiology domain, six factors in the implementation domain and three factors in the enabling environment domain (Table 2).

*Development*. Initial team discussions led to the identification of 25 factors considered important to assess country readiness for vivax elimination (S3 Table). Further discussions within the research team and consultations with NMP participants resulted in the refinement of this list, considering relevance to policy change. For example, the factor 'vulnerable population at risk' was deemed as less important by NMP participants given that policies are prepared to address the needs of all, including but not limited to vulnerable populations at risk.

The 14 remaining factors were validated with the NMP participants. NMP participant ranked factors such as *malaria program phase*, *vivax case load*, *G6PD prevalence*, *G6PD deficiency heterogeneity*, *liver stage treatment*, *antirelapse efficacy*, *patient adherence*, *human resources*, *and risk aversion* as having a high importance for policy making (S4 Table). NMPs also suggested additional factors including *vector control approach*, *insecticide resistance*, *quality of malaria commodities*, and *private sector availability*. However, those additional factors were ultimately not included as follow-up discussions with NMP participants revealed that those factors were deemed less relevant to policy changes and more relevant for programmatic decisions for radical cure.

Experts validated 13 out of 14 suggested factors by reaching agreement of over 85% on their importance for readiness and/or decision making on radical cure test and treat combinations during the first round of the modified e-Delphi process (S5 Table). The only factor for which experts did not reach immediate agreement (62% agreement) was '*political will*'. However, agreement was reached in a consecutive round (S6 Table).

**Table 2. The final BAT with 14 specific factors, their questions and categories.**

**Epidemiological domain**

| Specific factors | Question | Categories |
|---|---|---|
| 1. Phase of malaria program | What is the phase of malaria program in your country? | • Prevention of reintroduction (3 years of 0 locally acquired cases)<br>• Elimination (<1case/1000 population at risk per year)<br>• Pre-elimination (slide or RDT positivity rate <5%)<br>• Control (slide or RDT positivity rate ≥5%) |
| 2. Vivax case load | What is the number of annual reported cases of vivax in your country? | • 0<br>• 1–10,000<br>• >10,000 |
| 3. G6PD deficiency prevalence | What is the level of G6PD deficiency (defined as less than 30% G6PD activity) | • Rare (<1%)<br>• Common (1%-10%)<br>• High (>10%)<br>• Don't know |
| 4. Liver-stage treatment | What is the current radical cure regimen/s for uncomplicated vivax malaria recommend by the national treatment guideline in your country | • PQ14 days (0.25mg/kg/day for a total dose of 3.5mg/kg)<br>• PQ14 days (0.5 mg/kg/day for a total dose of 7mg/kg)<br>• PQ8 weekly (0.75mg/kg/week for a total dose of 6mg/kg)<br>• PQ7 days (0.5 mg/kg/day for a total dose of 3.5mg/kg) |
| 5. Anti-relapse efficacy | Anti-relapse efficacy data is available for which radical cure drug regimen/s in your country or similar settings? | • The efficacy of PQ14 low dose is estimated as **adequate** (≥85%) at six months. Risk of recurrence at 6 months is estimated as adequate at 1%, but >10% at 1 year.<br>• The efficacy of PQ14 low dose is estimated as **adequate** (≥85%) at six months. The risk of recurrence at 6 months in this scenario is estimated at around 10%<br>• The efficacy of PQ14 low dose is estimated as **inadequate (<85%).** The risk of recurrence at 6 months is estimated at around 20%<br>• The efficacy of PQ14 low dose is estimated as **inadequate (<85%).** The risk of recurrence at 6 months is estimated at around 40% |

**Implementation domain**

| Specific factors | Question | Categories |
|---|---|---|
| 6. Referral initiation and completion rate | What is the estimated proportion of vivax patients referred from initial point of malaria diagnosis to higher health centers? | • High (>80%)<br>• Moderate (>50%-80%)<br>• Low (>10–50%)<br>• Very low (<10%)<br>• Don't know |
|  | What is the estimated proportion of referred vivax patients that complete referral at receiving health facility? | • High (>80%)<br>• Moderate (>50%-80%)<br>• Low (>10–50%)<br>• Very low (<10%)<br>• Don't know |
| 7. Community-level case management | What activities are allowed by the Ministry of Health for health workers at the community level for malaria case management? | • HW can test, treat, and track patient adherence.<br>• HW can test and track but cannot treat.<br>• HW available but cannot test, treat, and track.<br>• HW not available at community level<br>• Don't know |
| 8. Health worker compliance rate | What do you think is the estimated proportion of health workers at different levels of the health system who adhere to current or new treatment protocols? | • High (>80%)<br>• Moderate (>50%-80%)<br>• Low (<50%)<br>• Don't know |
| 9. Patient adherence rate | What do you think is the estimated proportion of patients who adhere to the full treatment regimen of current recommended radical cure drugs? | • High (>80%)<br>• Moderate (>50%-80%)<br>• Low (<50%)<br>• Don't know |
| 10. Interventions to improve patient adherence | Is supervised treatment or any other intervention being implemented at a large scale to improve patient adherence to current recommended radical cure of vivax in your country? | • Yes<br>• No<br>• Don't know<br>• If yes, please mention the type of supervised treatment/ adherence intervention being implemented. |

*(Continued)*

**Table 2.** (Continued)

| 11. Pharmacovigilance | What is the status of adverse event reporting for any disease in the last 12 months in your country? | • High (AE usually recorded and reported from health facility to national level)<br>• Moderate (AE sometimes recorded and reported health facility to national level)<br>• Low (AE not recorded or reported health facility to national level)<br>• Don't know |
|---|---|---|
| **Enabling environment domain** | | |
| **Specific factors** | **Question** | **Categories** |
| 12. Budget | What percentage of the annual budget for malaria is funded by the national government? | • High (>90%)<br>• Moderate (10–90%)<br>• Low (<10%) |
| 13. Political will | Who was the chief guest in the last World Malaria Day event in your country? | • High (head of state like Prime minister attends the WMD*)<br>• Moderate (health/Permanent secretary attends WMD*)<br>• Low (No high-ranking official attends WMD*) |
| 14. Risk aversion of decision makers for future malaria policy options | What percentage of time was spent discussing "patient safety" compared to "efficacy" and "implementation issues" of 8-aminoquinolines (such as primaquine and tafenoquine) in the last Technical Working Group (TWG) meeting which discussed on treatment policy change for vivax malaria in your country? | • More time spent discussing patient safety<br>• Equal time spent discussing patient safety and efficacy & implementation issues.<br>• Less time spent discussing patient safety.<br>• Do not know / TWG meetings held only sporadically |

*World Malaria Day

Experts also suggested eight additional factors to the BAT which were *vivax case heterogeneity*, *severity of G6PD deficiency*, *safety of radical cure*, *feasibility of evidence use in policy change*, *variant type of enzyme CYP2D6 gene*, *insecticide resistance*, *quality of malaria commodities*, *existence of community outreach* (S7 Table). Out of the suggested additional factors, the study team identified the *severity of G6PD deficiency*, *safety of radical cure*, *feasibility of evidence use in policy change* as directly relevant for policy change. These three additional factors were added in round two of the first e-Delphi process (S7 Table). However, after consultation with NMPs participants and considering limitation of data availability in some country contexts, they were excluded from the final set of BAT factors.

**Questions to capture the factors.** To assess the 14 factors included in the BAT, questions to capture them were developed. For example, to capture the factor '*pharmacovigilance*' the question "*What is the status of adverse event reporting for any disease in the last 12 months in your country*?' was developed. The final set of questions can be found in Table 2.

*Development*. Through literature review and core research team discussion, an initial set of questions that would capture each factor was developed and NMPs participants considered all questions suitable. Experts validated the questions through the first and second modified e-Delphi process. Among the 14 questions only the question *"Who was the chief guest in the last World Malaria Day event in your country?"* to assess the factor '*political will*' did not reach agreement (67%) and comments suggested that the purpose of the proposed question was not clear (S8 Table). There was a discrepancy between experts and NMP participants as to the importance of asking about *political will* as part of enabling environment, with NMP participants prioritizing it.

**Categories for each factor.** To measure each factor, categories were designed. For example to measure the factor *vivax case load*, categories such as *0 cases*, *1–10,000 cases*, and *>10,000* cases were established [24, 25].

In the first round of the first modified e-Delphi process, experts agreed with 11 of the 14 categories. The categories for '*anti-relapse efficacy*', '*budget*' and '*political will*' did not receive agreement initially and required additional rounds to reach consensus (S9 Table). The categories for

'*anti-relapse efficacy*' designated as an appropriate threshold for adequate anti-relapse efficacy (defined as risk of recurrence and not risk/ probability of recurrence free) at six months reached agreement in round two with a threshold of 14.5% risk recurrence at 6 months deemed as effective treatment (range 50–100%). The categories for the additional factor '*severity of G6PD deficiency*' using the revised 'WHO classification of G6PD variants in homozygous and hemizygous individuals' [26], which was added only in round two, did not reach agreement (S10 Table).

## Scenarios for OAT

Eight scenarios were developed considering the final BAT factors and representativeness of different country scenarios across Asia Pacific region. Each scenario portrays a setting with distinct malaria epidemiology, health system, and enabling environment.

**Development.** During the initial development phase, five scenarios were developed which ranged from a scenario with no indigenous vivax cases, strong health system, and high-level political will, to a scenario with high vivax caseloads, weak health system, and low level of political will (S11 Table). Throughout an iterative process these initial scenarios were extended to eight scenarios considering different malaria program phases between phases of control, pre-elimination, and elimination, health system capacities and other factors included in BAT. Based in research team discussions three additional scenarios were added: an outbreak situation, a scenario with political instability, and one with limited availability of any data required for BAT. The details of those eleven scenarios are available in S12 Table. Each scenario was given a fictional country name from a fantasy name generator to avoid stigmatization and bias [27].

Based on the feedback from the initial round of the second modified e-Delphi exercise of scenario-based test and treat options, three pairs of scenarios were merged to create a final set of eight scenarios (S12 Table).

The final eight scenarios were validated with NMP participants with the aim of ensuring that they could recognize their own country context in one of the scenarios. In general, NMP participants felt that it was easy to choose a scenario that represented their context based on the epidemiological factors because of familiarity with the factor and available data for these factors. However, they faced challenges in identifying enabling factors (e.g., political will, risk aversion) due to limited data availability.

To give an illustrated example of the scenarios, the scenario Theuna-Floesal depicts countries in the Asia Pacific who are either sustaining prevention of re-introduction of malaria or aim for certification of malaria elimination by achieving zero cases for three consecutive years. While these countries have no indigenous malaria cases, they remain susceptible to outbreaks. These countries also have a high proportion of patient referral and completion rate at higher level facility for treatment; a high proportion of health workers compliance to treatment protocol; and a high rate of adherence to treatment among vivax patients. Domestic funding of NMP activities range from moderate (31–89%) to high ($\geq$ 90%) with likely availability of external technical assistance from donor agencies. There is a high political will to sustain elimination and prevent re-introduction of malaria, and a moderate level of risk aversion within the Ministry of Health and National Malaria Program (Fig 1).

The other seven scenarios are depicted in detail in supplemental files (S1–S7 Figs). An overview of the eight scenarios is provided in Table 3.

## Optimal test and treat options for each scenario

Experts were asked to match different G6PD testing (qualitative or quantitative) options (Table 4), along with blood-stage treatment and radical cure with single-dose tafenoquine and/ or different dosing of primaquine (Table 5) to the previously described scenarios.

**Scenario: THEUNA-FLOESAL**

**Epidemiological factors:**

**Malaria program phase**: The Theuna-Floesal countries are either sustaining prevention of reintroduction (PoR) or are in the process of certification of elimination, aiming for no cases for three consecutive years (Elimination phase). Despite their achievements, they are receptive to re-introduction.

**Vivax caseload**: The countries are characterized by zero indigenous vivax cases. However, they are susceptible to an outbreak due to an influx of migrants through a porous border. In the last month, up to 400 imported vivax cases have been detected.

**G6PD deficiency prevalence**: The G6PD deficiency prevalence is estimated as common (1-10%).

**Liver stage treatment:** The current recommended radical cure regime is PQ at a low dose (3.5mg/kg total dose) given over 14 days or weekly dose (0.75mg/kg) for 8 weeks.

**Anti-relapse efficacy**: The efficacy of the current PQ14 low dose regime is not known.

**Implementation factors:**

**Referral initiation rate**: A high proportion of vivax patients (i.e., >80%) get referred to a higher-level health facility after getting diagnosed at the community level.

**Referral completion rate**: A high proportion of referred vivax patients (i.e., >80%) avail treatment at a higher-level facility.

**Community-level case management:** The NMP has moved malaria case management to the health facility level as cases decreased. Thus, there are health workers in the community who can test to confirm malaria and track cases but cannot treat. Community levels activities include malaria awareness raising, surveillance, and referral of cases.

**Health worker compliance rate:** A high proportion of health workers (i.e., >80%) are estimated to comply with national malaria treatment protocols.

**Patient adherence rate**: A high proportion of vivax patients (i.e., >80%) are estimated to adhere to the recommended radical cure regimen.

**Interventions to improve patient adherence:** The NMP promotes hospitalization of malaria patients and provides supervised treatment like Directly Observed Therapy (DOT).

**Pharmacovigilance**: The capacity of pharmacovigilance is high in these countries, such that adverse events are usually recorded and reported from health facilities to the national level.

**Enabling factors:**

**Budget:** The proportion of NMP activities that are funded domestically ranges from moderate (31-89%) to high (≥ 90%). However, external technical assistance may be available from donor agencies.

**Political will:** There is a high political will to sustain elimination and prevent re-introduction e.g., the head of state like the Prime Minister attends the 'World Malaria Day' event in advocacy and commitment to sustain the achievements made.

**Risk aversion of decision makers for future malaria policy options:** There is a moderate risk aversion within the Ministry of Health and National Malaria Program. During NMP Technical Working Group (TWG) meetings, equal time is spent discussing 'patient safety' compared to 'efficacy' and 'implementation issues of 8-aminoquinolines'.

**Fig 1. Scenario description of THEUNA-FLOESAL.**

**Table 3. Overview of eight scenarios.**

| Domain/ factors | THEUNA-FLOESAL | CREOSO-OTROS | ACRINES | JOBLIL | PLOJI | GLAERA | ECHA-BLAOR | USPOS |
|---|---|---|---|---|---|---|---|---|
| **Epidemiological domain** | | | | | | | | |
| **Malaria program phase** | Prevention of re-introduction & Elimination | Elimination | Elimination | Elimination | Pre-elimination | Control | Control | Control |
| **Vivax caseload** | 0/ 1–10,000 | 1–10,000 | 1–10,000 | 1–10,000 | >10,000 | >10,000 | >10,000 | >10,000 |
| **G6PD Def. Prevalence** | Common | Common | High | Rare | Common–High | Common–High | Common—High | Common—High |
| **Current liver-stage treatment** | PQ14 (3.5mg/kg)/ PQ8Wk | PQ14 (3.5mg/kg) / PQ8Wk | PQ14 (3.5mg/kg) / PQ8Wk | PQ14 (3.5mg/kg) / PQ8Wk | PQ14 (3.5mg/kg)/ PQ8Wk | PQ14 (3.5mg/kg) / PQ8Wk | PQ14 (3.5mg/kg) /PQ8Wk | PQ14 (3.5mg/kg) /PQ8Wk |
| **Anti-relapse efficacy data** | No data available | Adequate (Risk of recurrence at 6 months at 1%, but >10% at 1 year) | Inadequate (The risk of recurrence at 6 months around 20%) | Inadequate (The risk of recurrence at 6 months around 20%) | Adequate (The risk of recurrence at 6 months around 10%) | Inadequate (The risk of recurrence at 6 months around 40%) | Adequate (The risk of recurrence at 6 months around 10%, but >10% at 1 year) / no data available | Inadequate (The risk of recurrence at 6 months around 40%) |
| **Implementation domain** | | | | | | | | |
| **Referral initiation rate** | High | Very low to moderate | Low to Moderate | Moderate | Low | Very low or don't know | Very low or don't know | Very low or don't know |
| **Referral completion rate** | High | High | High | Moderate | Low | Very low or don't know | Very low or don't know | Very low or don't know |
| **HW availability: community-level case management** | HW can test and track but cannot treat | HW can test and track but cannot treat | HW can test track but cannot treat | HW can test, treat, and track patient adherence | HW can test and track but cannot treat | HW can test and track but cannot treat | HW can test and track but cannot treat | HW can test and track but cannot treat |
| **HW compliance with protocols** | High | Moderate to High | Moderate to High | Moderate to High | Low or don't know | Low or don't know | Low or don't know | Low/don't know |
| **Supervised treatment for patient adherence** | Yes | Yes | Yes/No | Yes/No | Yes/No | No | No | No |
| **Patient adherence rate** | High | Low to moderate | Moderate | Moderate | Low or don't know | Low or don't know | Low or don't know | No or don't know |
| **Pharmacovigilance** | High | Moderate to high | Moderate to High | Low to Moderate | Low | Low | Low | Low |
| **Enabling environment domain** | | | | | | | | |
| **Budget (domestic funding)** | Moderate to High | Low to Moderate | Moderate | High | Low | Low to Moderate | Low to Moderate | Low |
| **Political will** | High | Moderate to High | Moderate to High | Moderate | Low to moderate | Moderate | Low | Low |
| **Risk aversion of decision makers for future malaria policy options** | Moderate | Moderate to high | Low | Moderate | High | High | Moderate-High /don't know | Don't know |

For each scenario, questions were asked assuming that either chloroquine (CQ) or artemisinin combination therapy (ACT) are used as first line schizonticidal treatment. For scenarios with ACTs, use of TQ was not provided as a treatment option, in line with the current TQ label [28].

Experts were initially asked to identify the most suitable G6PD test for each scenario. If the participants selected a qualitative test, a question was asked to ascertain the optimal treatment for both G6PD normal (≥30% G6PD enzyme activity) and deficient patients (< 30% activity). However, if a quantitative test was chosen, questions were posed to determine the optimal

**Table 4. G6PD testing options and their characteristics.**

| G6PD testing options | Type | Operational site |
|---|---|---|
| 1. Spectrophotometry test | Quantitative | Laboratory/hospital (not PoC) |
| 2. Fluorescent Spot Test (FST) | Qualitative | Laboratory/hospital (not PoC) |
| 3. Biosensor G6PD | Quantitative | PoC |
| 4. G6PD RDT* | Qualitative | PoC |

*The team was aware of the WHO notice of concern regarding the qualitative G6PD test. However, the option was retained in the Delphi exercise to get expert feedback on importance of this diagnostic option.

treatment for patients categorized as normal ($\geq$70% activity), intermediate (30–70% activity), and deficient (<30% activity) in G6PD enzyme levels.

For five out of eight scenarios, experts selected Point of Care (PoC) quantitative G6PD test to be the most suitable diagnostic (S13 Table). The majority of those five scenarios (*Creoso-Otros*, *Acrines*, *Joblil*) were placed in elimination settings (S1–S3 Figs). For three out of eight scenarios, a qualitative PoC test was selected (S13 Table). The majority of those three scenarios (*Echa-Blaor and Uspos*) were in control phase (S6 and S7 Figs).

When CQ was assumed to be the blood-stage treatment of choice, for scenario where quantitative testing was selected, TQ was suggested in seven out of eight scenarios as the preferred treatment for G6PD normal patients (agreement ranging between 36–57%), PQ14 (low dose) for intermediates in 6/8 scenarios (agreement ranging 33–62%) and weekly PQ for G6PD deficient patients in all scenarios (agreement 62–93%). When qualitative testing was selected, experts opted for PQ14 (low dose) for G6PD normal patients in six out of eight scenarios with agreement ranging between 43–59% and for G6PD deficient patients weekly PQ across all the scenario (agreement ranging between 62–100%) (S13 Table).

For settings where the blood-stage treatment was an ACT, TQ could not be selected. When preference was given to quantitative G6PD testing, experts suggested PQ7 (high dose) across all scenarios for G6PD normal patients, PQ14 (low dose) for G6PD intermediates and PQ8 weekly for G6PD deficient patients. When qualitative testing was selected, experts suggested using PQ14 (low dose) in G6PD normal patients among five out of eight scenarios and PQ8 weekly for G6PD deficiency (S13 Table).

Given the current unavailability of qualitative PoC G6PD testing, prior to NMP consultation this option was removed for presentation to NMPs (Table 6 and S13 Table). Experts were also asked about the importance of having a point of care qualitative diagnostics for G6PD available for the national malaria programs and 90% (18/20) responded positively. Independent of scenarios, experts suggested that when a qualitative diagnostic is available the most suitable treatment to be guided by this diagnostic would be PQ14 (low dose) (85%) and PQ7 (low dose) (70%).

**Table 5. Overview of different dosing of primaquine and tafenoquine.**

| Radical cure regimen | Total dose | Daily dose | Length |
|---|---|---|---|
| Low dose, long course Primaquine (PQ14- low dose) | 3.5mg/kg | 0.25mg/kg | 14 days |
| Low dose, short course Primaquine (PQ7- low dose) | 3.5mg/kg | 0.5mg/kg | 7 days |
| High dose, long course Primaquine (PQ14- high dose) | 7mg/kg | 0.5mg/kg | 14 days |
| High dose, short course Primaquine (PQ7- high dose) | 7mg/kg | 1mg/kg | 7 days |
| PQ weekly | 6mg/kg | 0.75mg/kg | 8 weeks |
| Single-dose Tafenoquine 300mg (adult) | 300mg fixed dose | | 1 day |

**Table 6. Scenario-based test and treat options (expert responses from modified e-Delphi when only *quantitative testing was available*).**

| Scenario | Optimal G6PD test | Liver-stage treatment option/s | | | | | |
|---|---|---|---|---|---|---|---|
| | | With CQ as the blood-stage treatment | | | With ACT as the blood-stage treatment | | |
| | | For G6PD Normal | For G6PD intermediate | For G6PD deficient | For G6PD Normal | For G6PD intermediate | For G6PD deficient |
| 1. Theuna-Floesal | Point of care Quantitative test for G6PD | TQ | PQ14 (low dose) | PQ8wkly | PQ7 (high dose) * | PQ14 (low dose) | PQ8wkly |
| 2. Creoso-Ortos | Point of care Quantitative test for G6PD | TQ | PQ14 (low dose) | PQ8wkly | PQ7 (high dose) * | PQ14 (low dose) | PQ8wkly |
| 3. Acrines | Point of care Quantitative test for G6PD | TQ | PQ14 (low dose)/ | PQ8wkly | PQ7 (high dose) * | PQ14 (low dose) / | PQ8wkly |
| 4. Joblil | Point of care Quantitative test for G6PD | TQ | PQ14 (low dose) | PQ8wkly | PQ7 (high dose) * | PQ14 (low dose) | PQ8wkly |
| 5. Ploji | Point of care Quantitative test for G6PD | TQ | PQ14 (low dose) | PQ8wkly | PQ7 (high dose) * | PQ14 (low dose) | PQ8wkly |
| 6. Glaera | Point of care Quantitative test for G6PD | TQ | PQ7 (low dose) | PQ8wkly | PQ7 (high dose) * | PQ14 (low dose) | PQ8wkly |
| 7. Echa-Blaor | Point of care Quantitative test for G6PD | PQ7 (high dose) * | PQ14 (low dose) | PQ8wkly | PQ7 (high dose) * | PQ14 (low dose) | PQ8wkly |
| 8. Uspos | Point of care Quantitative test for G6PD | TQ | PQ7 (low dose) | PQ8wkly | PQ7 (high dose) * | PQ14 (low dose) | PQ8wkly |

*Replaced by PQ14 (high dose) in G6PD normal patients when presented to NMP at the annual meeting in Dec 2022, given the issued WHO recommendation against high dose PQ7 in Nov 2022 [29].

In November 2022, when the NMP and expert consultations were ongoing, the WHO issued their revised treatment guidelines that included recommendation against PQ7 (high dose) [29]. In response to this, expert were asked to select the next best options if PQ7 (high dose) is not available and 64% opted for PQ14 (high dose) in G6PD normal patients. During the December 2022 piloting with NMPs that occurred after WHO recommendation, we therefore replaced PQ7 with PQ14.

**Approaches tools.** The 'approaches tools' aim to support NMP planning policy change and implementation steps consistent with the selected test and treat option based on scenarios and outlines different strategies to achieve high coverage with their choice of testing and treatment.

Specific approaches encompass targeted resource allocation, advocacy, testing and treatment strategies, improved accessibility, and adherence to radical cure. These approaches tools include tools that address policy considerations, such as assessing the needs for policy change, reviewing policy, and incorporating country specific evidence (S1 Text).

## Discussion

Vivax malaria poses a major challenge to the malaria elimination goal, requiring urgent improvements in current radical cure policy and practice. While the research community has developed multiple novel treatment regimens, deciding which options are the most feasible and impactful for a given context is complex.

The OAT aims to support NMPs to select the most suitable radical cure policy for their given scenario considering malaria epidemiology, health system, and politico-economic conditions and outline feasible approaches to policy change. The intention of the OAT is to guide the policy process and stimulate discussion rather than to prescribe solutions, users can therefore adapt and modify tools of the OAT to align with their specific context, fostering a sense of

ownership. Here we describe the major findings from the development of the tool. First experiences using the tool will be described elsewhere.

Vivax malaria in the Asia-Pacific region is heterogeneous and country contexts vary considerably. The a priori assumption in the development of the OAT was therefore that those different contexts matter for tailoring treatment policy. However the selected test and treat options varied little by scenario, suggesting that potentially a less differential approach to treatment policy is required.

In the OAT development process diverging views between experts and NMP participants were observed. For example, experts felt that the factor *'political will'* had little relevance for decision making, while NMP participants gave this factor greater importance. This could be attributed a more nuanced understanding of NMPs of the contexts within which they are required to make decisions. Technical experts and those focused on evidence informing policy making, may often overlook the important role of political system, institutional structures and political contestation in decision making [30].

Experts selected PQ7 (high dose) for one scenario (with CQ as blood-stage treatment) and across all scenarios (with ACT as blood-stage treatment) when quantitative G6PD testing is available (Table 6 and S13 Table). This is in contrast to the 2022 WHO recommendations against PQ7 (high dose) which cites concerns over gastro-intestinal tolerability and increased risk for hemolysis [29]. This discrepant interpretation of the available evidence is likely linked to varying levels of acceptance for risk and different sense of urgency [31]. Within the research community there is no consensus as to what constitutes an acceptable safety or tolerability profile for radical cure and how this can be balanced against increasing gains in efficacy to prevent relapsing malaria. Less than half of the included experts felt that a threshold of <1/100,000 hemolytic events requiring transfusion would be adequate to consider the treatment having an appropriate safety profile, compared to 38% suggesting <1/10,000 and 15% suggesting <1/1000 to be acceptable, with most experts mentioning dependency on contextual factors such as health system capacity (S8 Table). Defining acceptable tolerability of non-life-threatening events such as gastro-intestinal disturbance is even more difficult. More recent data however further support the use of higher primaquine doses in settings where G6PD testing can be provided [14, 32, 33].

Similarly, there is a lack of consensus on what constitutes effective radical cure. For falciparum malaria the established threshold is 95% efficacy at day 28 and therapeutic efficacy studies are conducted regularly providing data that then support a change in first line treatment should [34]. Within this study, experts suggested that 85% efficacy, or in other words a 15% risk of recurrence at 6 months to be an adequate threshold for efficacy of radical cure drug regimen. To put this in context, in a recent individual patient data meta-analysis the risk of recurrence with low dose primaquine at 6 months was 19.3%, with 22% and 17% in low and high relapse areas, respectively [13]. Similarly, the risk of recurrent *P. vivax* malaria at 6 months for tafenoquine ranges from 28% to 38% [15, 16].

Although the qualitative G6PD test is currently not available due to the WHO's notice of concern [35], feedback from experts within the Delphi process highlighted that there remains a rationale for the use of qualitative tests in certain scenarios such as resource limited settings with limited capacity for training of health care workers, low G6PD prevalence [36].

Toolkits can be an effective way to navigate complexities of policy making, particularly in cases where there is limited evidence available [37]. Several toolkits have been developed to support decision making at policy levels. For example, the SUPPORT Tools for evidence-informed health Policymaking (STP) developed to aid the decision-makers in health policies and programs is comprehensive and includes aspects ranging from the synthesis of evidence to its application in the policy making process [38]. Similar to the OAT, it presents scenario-

based questions, that assist policymakers in recognizing issues and defining their attributes. However, policy making still requires tailored strategies for the country context. Similarly, the READ approach (Ready for materials, Extract data, Analyze data, Distil your findings) uses rigorous document analysis to understand policy contents, processes and policy related discussions. It provides guidance using a systematic approach for content analysis in relevant documents for health policy broadly, encompassing how to ready the materials, extract data, analyze data, and distil findings as applied in research papers [39]. Nonetheless, none of these offer a decision support toolkit for policy making specific to vivax malaria. The OAT advances the existing decision-making tools as it incorporates all key factors critical for radical cure policy change, which is particularly relevant when multiple options are available.

Research indicates that providing decision-makers with multiple options to choose from, can lead to delays in the decision-making process. For instance in one study on HIV antiretroviral treatment prioritization, investigators showed that differing motivation and underlying ethical considerations can lead to disagreements about priority settings and hence delay the decision making process [20]. In the wider literature, the presence of multiple options has been found to be counterproductive particularly because they could add complexity to decision making [40]. The presence of multiple options means a need to review them carefully against each other and thus can be resource intensive and have been proven to engender conflicts and delay in decision making [41]. Similar challenges can arise in the context of malaria, where NMPs are likely to face dilemmas when selecting the optimal approach for revising their policies on vivax radical cure [19]. Reducing delays in decision making processes is key to accelerate access to tools that can support NMP goals of malaria elimination, which most countries in the Asia Pacific region aim to do by 2030.

There are several limitations to this work. Firstly, the OAT is designed as a tool to guide discussion of and with decision makers, rather than as a tool kit where variables are entered and a specific output recommendation for optimal test and treatment options generated. Secondly, experts evaluated evidence available at the time, but research is a dynamically developing field. There is a need to iteratively update the OAT to reflect current evidence for best practice. How this can be done efficiently is subject to further work. Third, the Delphi process was lengthy and demanding, which may have resulted in experts providing responses without the same level of careful consideration that they might have given in a shorter format. Fourth, the toolkit requires tailoring to the political and economic context of the implementing country [42]. While the NMP participants reviewed the eight different scenarios and felt they adequately represented countries across the region, the scenarios do not specifically represent individual country context. Lastly, we were not able to assess the ease of use and potential impact of OAT use on the policy making process. This is, however, subject to further work.

## Conclusions

We developed a toolkit to assist NMP in evidence decision-making considering health system context and enabling environment. The OAT toolkit has the potential to assist in the development of policies for vivax radical cure, tailored to the unique context of each country, involving NMPs and stakeholders. Subsequent work is required to examine the effectiveness and feasibility of using the OAT.

## Supporting information

**S1 Table. Demographic characteristics of responding experts for BAT, modified e-Delphi.** (PDF)

**S2 Table. Comparison of originally intended composition of toolkit and final composition.**
(PDF)

**S3 Table. The 25 factors initially included in the BAT and the reasons for their inclusion or exclusion.**
(PDF)

**S4 Table. Ranking of factors included in the BAT by the NMP participants.**
(PDF)

**S5 Table. Results from round one of the first modified e-Delphi to validate the factors included in the BAT.**
(PDF)

**S6 Table. Response on the factors in round two of the first modified e-Delphi.**
(PDF)

**S7 Table. Additional factors suggested by experts.**
(PDF)

**S8 Table. Responses on the questions in each factor/ additional factor BAT (Round two of the first modified e-Delphi.**
(PDF)

**S9 Table. Detailed responses on the questions and categorizations in each factor BAT (Round one of the first modified e-Delphi).**
(PDF)

**S10 Table. Responses on the categorizations in each factor/ additional factor BAT (Round two Delphi).**
(PDF)

**S11 Table. Overview of scenarios using a simplified version of BAT.**
(PDF)

**S12 Table. Overview of 11 scenarios.**
(PDF)

**S13 Table. Scenario-based test and treat options (expert agreement on responses from Delphi- Qualitative and quantitative G6PD testing).**
(PDF)

**S1 Fig. Scenario CREOSO-OTROS.**
(PDF)

**S2 Fig. Scenario ACRINES.**
(PDF)

**S3 Fig. Scenario JOBLIL.**
(PDF)

**S4 Fig. Scenario PLOJI.**
(PDF)

**S5 Fig. Scenario GLAERA.**
(PDF)

**S6 Fig. Scenario ECHA-BLAOR.**
(PDF)

**S7 Fig. Scenario USPOS.**
(PDF)

**S1 Text. Approaches tool for OAT.**
(PDF)

**S2 Text. Step-by-step guide on using the OAT.**
(PDF)

## Acknowledgments

We would like to express our gratitude to the participants of the annual meeting of the Asia Pacific Malaria Elimination Network (APMEN) Vivax Working Group in 2022 in Bangkok for providing initial feedback to the toolkit development.

## Author Contributions

**Conceptualization:** Sanjaya Acharya, Manash Shrestha, Josselyn Neukom, Karma Lhazeen, Caroline A. Lynch, Kamala Thriemer.

**Formal analysis:** Sanjaya Acharya, Manash Shrestha, Josselyn Neukom, Caroline A. Lynch, Kamala Thriemer.

**Funding acquisition:** Caroline A. Lynch.

**Investigation:** Sanjaya Acharya, Ngo Duc Thang, Lyndes Wini, M. Naeem Habib, Josselyn Neukom, Karma Lhazeen.

**Methodology:** Sanjaya Acharya, Manash Shrestha, Josselyn Neukom, Caroline A. Lynch, Kamala Thriemer.

**Project administration:** Manash Shrestha.

**Supervision:** Caroline A. Lynch, Kamala Thriemer.

**Writing – original draft:** Sanjaya Acharya.

**Writing – review & editing:** Manash Shrestha, Ngo Duc Thang, Lyndes Wini, M. Naeem Habib, Josselyn Neukom, Karma Lhazeen, Caroline A. Lynch, Kamala Thriemer.

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
