## [Decision Letter · Decision Letter 0]

2 Apr 2024

PGPH-D-24-00225

Optimizing test and treat options for vivax malaria: an options assessment toolkit (OAT) for Asia Pacific National Malaria Control Programs

Dear Dr. Thriemer,

Thank you for submitting your manuscript to PLOS Global Public Health. After careful consideration, we feel that it has merit but does not fully meet PLOS Global Public Health’s publication criteria as it currently stands. Therefore, we invite you to submit a revised version of the manuscript that addresses the points raised during the review process.

We look forward to receiving your revised manuscript.

Kind regards,

Ruth Ashton, Ph.D.

Academic Editor

Journal Requirements:

2. Please ensure that Funding Information and Financial Disclosure Statement are matched.

3. Please provide separate figure files in .tif or .eps format only and remove any figures embedded in your manuscript file. Please also ensure all files are under our size limit of 10MB.

4. We have noticed that you have uploaded Supporting Information files, but you have not included a list of legends. Please add a full list of legends for your Supporting Information files after the references list.

Additional Editor Comments (if provided):

Reviewers' comments:

Reviewer's Responses to Questions

**Comments to the Author**

1. Does this manuscript meet PLOS Global Public Health’s publication criteria? Is the manuscript technically sound, and do the data support the conclusions? The manuscript must describe methodologically and ethically rigorous research with conclusions that are appropriately drawn based on the data presented.

Reviewer #1: Partly

Reviewer #2: Partly

2. Has the statistical analysis been performed appropriately and rigorously?

Reviewer #1: N/A

Reviewer #2: N/A

3. Have the authors made all data underlying the findings in their manuscript fully available (please refer to the Data Availability Statement at the start of the manuscript PDF file)?

Reviewer #1: Yes

Reviewer #2: No

4. Is the manuscript presented in an intelligible fashion and written in standard English?

Reviewer #1: Yes

Reviewer #2: Yes

5. Review Comments to the Author

Reviewer #1: In this work, Acharya and colleagues describe a toolkit to support national malaria programs in the Asia Pacific region in the choice of their P. vivax radical cure strategy. The toolkit consists in a list of questions used to assess the situation and classify any given country into one of eight archetypical scenarios. Each scenario is then associated with a recommended radical cure policy. The elaboration of the toolkit was the result of consultations with NMP members and a panel of experts (protocol detailed in a previous publication). Although the toolkit was presented to relevant stakeholders, the authors indicate that the toolkit has not yet been utilized in a specific country context.

The paper is very interesting to read and it tackles an important topic, namely supporting decision making in a situation where the recommendations and types of diagnostics and treatments are difficult to navigate. I highly appreciate the inclusion of various stakeholders with diverse range of expertise and origin in the conception of the tool. Nonetheless, I have some major comments that I believe the authors should address for publication, as well as some minor remarks.

Major point

1- In my opinion, the link between the eight country scenarios and their associated recommended test and treat policy is unclear. The manuscript describes in details how these choices were made as a result of expert consensus, but does not justify why such policies are the most appropriate for each scenario given the current evidence. I believe such a justification should be included in the manuscript (and potentially the toolkit), so that the recommendations reflect the results of evidence-based decision making, beyond expert opinions. This is particularly important, especially as the authors mention that some recommendations from the experts are not aligned with those from the WHO (see discussion section).

2- It seems that the test and treat recommendations for scenarios 1 to 5 are identical (and also scenarios 6 and 8). It would be useful if the authors could explain the need to have so many scenarios if they lead to the same policy.

3- I found it difficult to reconcile the objective of creating a user-friendly tool while stating that it should not be used as a prescriptive tool (in Discussion section: “a tool to guide discussion of and with decision makers, rather than as a tool kit where variables are entered and a specific output recommendation for optimal test and treatment options generated.”) It would be helpful if the authors could explain more precisely what makes their toolkit a support for discussion rather than a prescriptive tool.

4- The authors mention the need to constantly update the toolkit to adapt to new research evidence. How should this be undertaken? Should the whole consultation process outlined in the manuscript be reproduced? It would be useful if the authors could provide more details as to how they envisage this.

5- The limitation section could be expanded. Specifically, a critical assessment of the limitations of the consultation and Delphi approaches would need to be outlined.

Minor points

6- I seems there is a typo in a table2, line 3, middle column: the text is duplicated from the previous row.

7- I did not understand the purpose of the approaches tool: I would invite the authors to reformulate the corresponding section for better clarity.

Reviewer #2: It would interesting to understand why only 3 NMP contributed to this exercise? how representative are they really? Likewise it would interesting to know on what criteria were the so-called "experts" selected? The fact that some of the "options" such as the high-dose primaquine short regimen, were removed, despite the evidence that it is safe, it problematic. Would be also very interesting to know the percentage of the NMP people who really understand P.vivax malaria. And what proportion actually have access to the scientific evidence?

6. PLOS authors have the option to publish the peer review history of their article (what does this mean?). If published, this will include your full peer review and any attached files.

**Do you want your identity to be public for this peer review?** For information about this choice, including consent withdrawal, please see our Privacy Policy.

Reviewer #1: No

Reviewer #2: No

---

## [Editor Report · Decision Letter 1]

24 Apr 2024

Optimizing test and treat options for vivax malaria: an options assessment toolkit (OAT) for Asia Pacific National Malaria Control Programs

PGPH-D-24-00225R1

Dear Dr. Thriemer,

We are pleased to inform you that your manuscript 'Optimizing test and treat options for vivax malaria: an options assessment toolkit (OAT) for Asia Pacific National Malaria Control Programs' has been provisionally accepted for publication in PLOS Global Public Health.

Best regards,

Ruth Ashton, Ph.D.

Academic Editor